# A Scoping Review and Meta-Analysis of Anti-CGRP Monoclonal Antibodies: Predicting Response

**DOI:** 10.3390/ph16070934

**Published:** 2023-06-27

**Authors:** Ja Bin Hong, Kristin Sophie Lange, Lucas Hendrik Overeem, Paul Triller, Bianca Raffaelli, Uwe Reuter

**Affiliations:** 1Department of Neurology, Charité Universitätsmedizin Berlin, Corporate Member of Humboldt University and Free University Berlin, 10117 Berlin, Germany; 2Doctoral Program, International Graduate Program Medical Neurosciences, Humboldt Graduate School, 10117 Berlin, Germany; 3Clinician Scientist Program, Berlin Institute of Health at Charité (BIH), 10117 Berlin, Germany; 4Universitätsmedizin Greifswald, 17475 Greifswald, Germany

**Keywords:** calcitonin gene-related peptide receptor antagonists, antibodies, monoclonal, migraine disorders, migraine without aura, migraine with aura, migraine disorders/prevention and control, migraine disorders/drug therapy, projections and predictions, treatment outcome, humans

## Abstract

Calcitonin gene-related peptide-targeted monoclonal antibodies (CGRP mAbs) are increasingly being used as preventive treatments for migraine. Their effectiveness and safety were established through numerous randomized placebo-controlled trials and real-world studies, yet a significant proportion of patients do not respond to this treatment, and currently, there is a lack of accepted predictors of response to guide expectations, as data from studies so far are lacking and inconsistent. We searched Embase and MEDLINE databases for studies reporting on predictors of response to CGRP and/or CGRP-receptor (CGRP-R) mAbs, defined as a 30% or 50% reduction in monthly headache or migraine days at varying durations of follow-up. Quantitative synthesis was performed where applicable. We found 38 real-world studies that investigated the association between various predictors and response rates. Based on these studies, good response to triptans and unilateral pain with or without unilateral autonomic symptoms are predictors of a good response to CGRP(-R) mAbs. Conversely, obesity, interictal allodynia, the presence of daily headaches, a higher number of non-successful previous prophylactic medications, and psychiatric comorbidities including depression are predictive of a poor response to CGRP(-R) mAbs. Future studies should confirm these results and help to generate more tailored treatment strategies in patients with migraine.

## 1. Introduction

Migraine is a debilitating neurological disorder that affects more than one billion people worldwide. It is characterized by severe headache attacks, often accompanied by vegetative symptoms such as nausea, vomiting, photo-, phono-, or osmophobia, and a broad spectrum of physical, mental, and psychological symptoms, varying throughout different phases of a migraine attack [1]. This highly disabling headache disorder incapacitates patients for 4–72 h during attacks and has a major impact on patients’ daily life and society [2]. The traditional first-line preventative drugs for migraine include beta blockers, antidepressants, and anticonvulsants, but lack specificity as they were originally developed for alternative purposes [3]. Consequently, their effectiveness, tolerability, and adherence can be inadequate and unpredictable [4].

In 1990, calcitonin gene-related peptide (CGRP) was identified as a pivotal neuropeptide released during the headache phase of migraine attacks [5]. CGRP is a highly potent vasodilating messenger peptide that is primarily released from sensory nerves and is involved in pain signaling and inflammation pathways [6]. This finding led to the development of CGRP-targeted monoclonal antibody (mAb) and small molecule therapies as new classes of medications for the treatment of migraine [7].

The monoclonal antibodies targeting CGRP (galcanezumab, fremanezumab, and eptinezumab) or the CGRP receptor (-R, erenumab) were shown to be safe and effective in the preventive treatment for episodic (EM) and chronic migraine (CM), even in patients who did not respond to numerous oral first-line treatment options [8,9,10]. The CGRP(-R) mAbs are larger molecules with limited ability to cross the blood–brain barrier (BBB). As such, they are thought to interfere with CGRP-signaling, primarily in the peripheral nervous system, in blood vessels and trigeminal afferent fibers in the meninges [11], and neuronal cell bodies and glial cells in the trigeminal ganglion, which is not protected by the BBB [12,13]. They have several advantages over traditional migraine preventive medications: a longer half-life means monthly or quarterly administrations are sufficient, a metabolism through general proteolytic degradation means pharmacokinetic interactions with other medications are unlikely [14], and an excellent tolerability profile contributes to a superior patient adherence [15].

The efficacy of CGRP(-R) mAbs was robustly demonstrated across studies conducted thus far [16], including randomized-clinical trials [8,17,18,19,20,21,22,23,24,25,26], open-label extension studies [27,28,29,30,31,32], and real-world observational cohort studies [33]. The majority of reported ≥50% responder rates ranged approximately from 40% to 70%. The benefit of CGRP(-R) mAb was seen regardless of migraine type (episodic or chronic) or ethnicity of the cohort. CGRP(-R) mAbs were just as efficacious for menstrual-related migraine, with reported ≥50% responder rates of 49.4% [34] and 57.5% [35]. Even though headaches did occur more frequently on menstrual days even in responders to erenumab, they tended to respond better to acute medication than in nonresponders [36]. Up to 40% of patients show an exceptionally high response (≥75% reduction in monthly headache days) or even reach complete migraine freedom [37].

Despite the generally favorable results of CGRP(-R) mAbs in migraine, up to one-third of patients do not benefit from CGRP(-R) mAbs, showing no change or even an increase in migraine frequency [33,38]. Those who then embark on a treatment trial with a different CGRP(-R) mAb after one treatment failure are less likely to respond to the second CGRP(-R) mAb [39,40]. The lack of response to a novel treatment that is touted by some as life-changing often begins after failures to multiple previous prophylactic attempts and can come as a severe disappointment to affected patients. When counseling patients on the possible treatment with CGRP(-R) mAbs, it is of critical importance to correctly inform them not only of the risk of side effects, but also of the expected benefit of treatment to the best of our knowledge. Currently, clinicians can only provide a general likelihood of treatment success based on reported overall response rates.

With knowledge about predictors of treatment success, clinicians could give more tailored and accurate advice to each individual patient. Knowing which factors are associated with a better response may also help guide decisions on whether to continue treatment at the 3-month juncture, which is when the desired effect is not evident yet, and continuation for an additional 3 months may or may not achieve the treatment goal. The knowledge of variables that explain a treatment failure may provide insight into the underlying mechanisms and aid in the design of effective therapies for individuals who do not benefit from CGRP(-R) mAbs.

This paper aims to review and analyze the current literature on predictors of response to CGRP-targeted therapies in real-world settings for migraine treatment. To our knowledge, a review on this topic is not yet published. We will examine available data for potential factors and discuss the possible implications for clinical practice and future research directions. Ultimately, a better understanding of the factors that influence response to CGRP therapies could lead to more personalized and effective treatment for individuals with migraine.

## 2. Results

After screening 744 titles and abstracts from Embase and 572 titles and abstracts from MEDLINE, 52 full texts were retrieved. Fourteen studies were excluded either because they were based on the same cohort as an included study or because they did not provide measures of the association of interest. We included 38 studies [41,42,43,44,45,46,47,48,49,50,51,52,53,54,55,56,57,58,59,60,61,62,63,64,65,66,67,68,69,70,71,72,73,74,75,76,77,78] in this review, which are summarized in Appendix A. The PRISMA flow diagram of the literature search process is displayed in Figure 1.

### 2.1. Demographic Characteristics

We found three studies [70,71,72] specifically designed to determine if specific demographic factors predict response to CGRP(-R) mAbs. Cetta et al. showed that similar reductions in monthly headache days (MHD) were seen in patients under and over the age of 65 [71]. Guerzoni et al. showed no difference in mean MHD reduction between patients pre- and post-menopause [72]. In the ESTEEMen study by Ornello et al., a pooled patient-level analysis of 1410 migraine patients, response to erenumab did not differ between male and female patients [70]. Other studies examined existing cohorts’ response rates and their correlation with demographic factors. Overall, the results from published studies do not support the hypothesis that age or sex determine the response to CGRP(-R) mAbs. Odds ratios (OR) and 95% confidence intervals (CI) for the factors of age, sex, and other baseline demographic variables from included studies are shown in Figure 2 and Figure 3. Lekontseva et al. found being employed as opposed to being unemployed increased the odds of being a responder (OR: 3.82, 95% CI: 1.36–10.73) [62]. A negative family history of migraine was associated with a decreased likelihood of response to CGRP(-R) mAbs (OR: 0.4, 95% CI: 0.16–0.97) in one study [42].

#### Weight

Barbanti et al. [51] found obesity, defined as a body mass index (BMI) of ≥30 kg/m^2^, to be a negative predictor of response to CGRP(-R) mAbs in patients with CM (OR when compared to patients with normal weight: 0.21, 95% CI: 0.07–0.63). A negative association between obesity (BMI ≥ 30 kg/m^2^) and treatment response was also seen in Salem-Abdou et al. [66], though a statistical significance in multivariable analysis was not reached (OR: 0.57, 95% CI: 0.29–1.13). Higher BMI in kg/m^2^ as a continuous variable tended to lead to a slight decrease in the likelihood of responding to CGRP(-R) mAbs (Figure 4).

### 2.2. Migraine Attack Features

Overall, the presence of symptoms that are characteristic of migraine attacks, such as unilateral pain or accompanying photo-/phonophobia, nausea, and vomiting, tended to predict a better response to CGRP(-R) mAbs.

#### 2.2.1. Unilateral Pain

Three studies [42,48,55] found that having unilateral headache predicted a better response to CGRP(-R) mAbs (Figure 5). In Raffaelli et al. [55], super-responders (with ≥75% reduction in MHDs) were more likely to have unilateral pain than non-responders (≤25% reduction in MHDs), although the difference did not reach statistical significance (crude OR: 4.51, 95% CI: 0.72–28.32). Vernieri et al. [48] found unilateral pain more frequent than bilateral pain in the group of 3-month responders, with a significant difference in the mean reduction in MHDs (median of −13 days with interquartile range of 9.5 days in the unilateral vs. median of −5.5 days with interquartile range of 12.5 days in the bilateral group, *p* = 0.004). Nowaczewska et al. [42] observed that for both unilateral fixed and unilateral alternating pain, the likelihood of responding to CGRP(-R) mAbs was higher, independent of other variables that were also associated with ≥50% response. Ihara et al. reported a nonsignificant univariable OR for unilateral pain and response rate (0.79, 95% CI: 0.33–1.89) [56].

#### 2.2.2. Presence of Accompanying Symptoms

Barbanti et al. [51] saw that the presence of both unilateral pain and unilateral autonomic symptoms was associated with an increased response rate to CGRP(-R) mAbs, both in patients with high-frequency episodic migraine (HFEM) and CM. The combination of unilateral pain and allodynia predicted a ≥50% response in patients with CM (OR: 1.71, 95% CI: 1.04–2.82). In De Matteis et al. [76] patients with cranial autonomic symptoms had a higher reduction in MHDs at 12 weeks than those without cranial autonomic symptoms (median −10 days, interquartile range [IQR] −15 to −6 vs. median −6 days, IQR −12 to −3, *p* = 0.009). The presence of vomiting during migraine attacks was positively associated with a ≥ 75% reduction in MHD after 3 months of treatment with CGRP(-R) mAbs (crude OR: 3.952, 95% CI: 1.08–17.417) [55]. Patients with moderate photophobia as opposed to no photophobia in their headaches tended to respond better to CGRP(-R) mAbs, though statistical significance was not reached in multivariable analysis [56]. In Lee et al. [60] the absence of all accompanying symptoms, defined as nausea, vomiting, photophobia, and phonophobia during headache attacks was predictive of a nonresponse to CGRP(-R) mAbs (OR: 0.314, 95% CI: 0.118–0.834). There was no difference in response rates between migraine with and without aura [69].

#### 2.2.3. Sensitization

The presence of sensitization to stimuli, even during interictal states at baseline, was highly predictive of nonresponse to CGRP(-R) mAbs. Ashina et al. [43] performed quantitative sensory testing during a non-ictal phase before the administration of CGRP(-R) mAbs and found that a pathologically heightened sensitivity that fulfilled the definition of allodynia was associated with a less than 50% reduction in monthly migraine days (MMD) after 3 months of treatment. This association was especially evident in cephalic allodynia (OR for ≥50% response: 0.037, 95% CI: 0.004–0.181), but was also seen with extra-cephalic allodynia (OR for ≥50% response: 0.188, 95% CI: 0.022–0.954). A higher heat pain threshold, which represents a lower sensitivity to thermal stimuli and predicted a good response to CGRP(-R) mAbs (OR for higher heat pain thresholds in °C: 2.57, 95% CI: 1.08–6.11), was found in another study [44]. In Pensato et al. [68], the presence of allodynia as a clinical symptom was associated with nonresponse after 3 months of treatment with CGRP(-R) mAbs (OR for ≥50% response: 0.47, 95% CI: 0.24–0.94).

### 2.3. Migraine History

Two predictors of response to CGRP(-R) mAbs related to migraine history stood out upon analyzing published evidence: (a) the daily presence of headaches in chronic migraine, which was associated with nonresponse, and (b) response to triptans in the treatment of an acute migraine attack, which predicted a better response (Figure 6).

#### 2.3.1. Continuous Chronic Migraine and Monthly Headache Days at Baseline

Migraine patients having daily headaches, described as continuous CM in some papers, were overrepresented in the group of nonresponders to CGRP(-R) mAbs. Figure 6 shows a quantitative summary of odds ratios from four studies, showing a pooled effect size of 0.24 (95% CI: 0.14–0.43). In Lowe et al. [61], migraine patients with daily headaches made up 86.7% of all nonresponders to erenumab. In Schoenen et al. [53], 86.7% of patients with daily headaches were nonresponders. The number of monthly headache or migraine days at baseline alone did not show a significant relationship with response rate (Figure 7).

#### 2.3.2. Migraine Burden at Baseline

Scores reflecting migraine burden, such as Headache Impact Test-6 (HIT-6) and Migraine Disability Assessment (MIDAS) measured at baseline, displayed a less clear relationship with response to CGRP(-R) mAbs, with most odds ratios not reaching significance (Figure 7).

#### 2.3.3. Response to Triptans

A favorable response to triptan acute medication was associated with a good response to CGRP(-R) mAbs. Most studies reported odds ratios between 2 and 6 that, when pooled, yielded an overall OR of 2.66 (95% CI: 1.73–4.09, Figure 6). During a migraine attack, response to triptans leads to a decrease in CGRP levels, and migraine headaches that respond to triptans had higher levels of CGRP [79]. Frattale et al. [52] hypothesize that in patients with good triptan response, CGRP is the main mediator in generating and maintaining migraine attacks, and targeting CGRP signaling is effective in preventing migraines, while in triptan non-responders, other mediators of pain not addressed by CGRP(-R) mAbs may play a bigger role.

#### 2.3.4. Medication Overuse

Most studies reported odds ratios for medication overuse (MO) at baseline and ≥50% responses that were not significant. One study [48] saw that the presence of MO at baseline was predictive of a ≥50% response in MHD to CGRP(-R) mAbs (OR: 4.58, 95% CI: 1.49–14.06); this association was not found in other studies. In most studies, the presence of MO at baseline decreased the likelihood of achieving ≥50% response (Figure 8).

#### 2.3.5. Previous Preventive Treatments

Most patients included in real-world studies of CGRP(-R) mAbs already failed multiple classes of prophylactic medications, as most health insurance providers do not yet cover the costs of CGRP(-R) mAbs otherwise. Consequently, the reported median number of previous medication failures in study cohorts ranged from 4 to 7, representing a therapy-resistant or difficult-to-treat subset of migraine patients. Within this population, there was a small but consistent negative association between the number of previous preventive medication failures and response rate to CGRP(-R) mAbs (pooled OR: 0.83, 95% CI: 0.73–0.93, Figure 8). The more non-successful treatment attempts patients had to different classes of migraine preventives, the less likely they were to respond to CGRP(-R) mAbs.

### 2.4. Comorbidities

#### 2.4.1. Psychiatric Comorbidities

Patients suffering from migraine are about two to five times more likely to suffer from depression or anxiety disorders [80]. In real-life studies of CGRP(-R) mAbs, patients with psychiatric comorbidities in general, and comorbid depression in particular, were less likely to be responders (pooled OR for psychiatric comorbidity: 0.56, 95% CI: 0.33–0.94, for comorbid depression: 0.51, 95% CI: 0.35–0.73, Figure 9). Associations were also seen between response failure and traits imbuing psychological vulnerability, such as personality disorders and the number of serious stressful events [45] (Figure 9). Lovati et al. [46] found non-responders (<30% reduction in MHD) scored higher on disinhibition, anhedonia, depressivity, and distractibility on the Personality Inventory for DSM-5 (PID-5) questionnaire than full-responders (>50% reduction in MHD) [46]. In Driessen et al., however, the response rate to fremanezumab was not significantly different in subgroups with major depressive disorder or generalized anxiety disorder [47].

There was significant variability in the methods used to determine comorbid psychiatric conditions. Most studies used the data collected at baseline on medical history and comorbidities, either through electronic chart review [47,55,56,66,68] or baseline in-person interviews [64,74]. In Bottiroli et al. [45], a complete psychological evaluation was performed at baseline using the Structured Clinical Interview for DSM-5, Clinical Version (SCID-5-CV) [81] by two expert psychologists, and a series of self-filled questionnaires were administered, including the Hospital Anxiety and Depression Scale (HADS) [82], Childhood Trauma Questionnaire, and the Stressful Life-events Questionnaire [83]. Lee et al. [63] assessed for depressive and anxiety symptoms using the Patient Health Questionnaire-9 (PHQ-9) [84] and the General Anxiety Disorder-7 (GAD-7) [85] and defined the presence of depression or anxiety using a cutoff value. Lovati et al. [46] evaluated personality traits using the 220-item Personality Inventory for DSM-5 (PID-5) [86].

#### 2.4.2. Other Comorbidities

Few studies reported on the effect of non-psychiatric comorbid conditions on treatment response. Vernieri et al. [48] found comorbid gastroesophageal reflux disease indicates a predisposition to treatment failure (OR: 0.175, 95% CI: 0.047–0.659). Ihara et al. [56] found that a history of immuno-rheumatological disorders was linked to treatment failure (OR: 0.027, 95% CI: 0.002–0.422). These results, while giving rise to interesting hypotheses, need validation through larger cohort studies.

### 2.5. Other Predictors

One study [42] examined the maximum flow velocity in the middle cerebral artery (MCA) using transcranial Doppler in the absence of comorbid intra- or extracranial stenosis and found non-responders to erenumab and fremanezumab had higher mean velocities in both MCAs at baseline compared to good responders (73.14, standard deviation [SD] 12.96 cm/s and 73.16, SD 15.62 cm/s in non-responders vs. 64.22, SD 15.17 cm/s and 65.78, SD 13.37 cm/s in good responders, *p* = 0.001/0.0073). The authors discussed two possible explanations for higher peak velocities: increased cerebral blood flow in the anterior circulation or a decreased vessel lumen through less dilation [42].

We found two studies [41,77] examining baseline CGRP levels as predictors of response. Alpuente et al. measured baseline salivary CGRP levels before erenumab treatment and found that higher levels were independently associated with treatment response in patients with EM (OR: 1.09, 95% CI: 1.01–1.10) [41]. De Vries Lentsch et al. [77], on the other hand, did not find an association between serum CGRP-like immunoreactivity at baseline and reduction in MMD (*P* = 0.24). Zecca et al. looked at 15 common single nucleotide polymorphisms (SNPs) of CALCRL (calcitonin receptor like receptor) and RAMP1 (receptor activity modifying protein 1) genes and their association with ≥50% responder status [57]. One genetic variant, RAMP1 rs7590387, was less frequent in ≥75% responders compared to non-responders (OR per G allele: 0.53, 95% CI: 0.29–0.99) though statistical significance was lost with multivariable analysis. These results require validation through larger studies.

## 3. Discussion

With a systematic search of predictors of response to CGRP(-R) mAbs, we were able to identify a group of studies that, when analyzed together, revealed a number of consistent associations that were hitherto not well known. Combining data from multiple smaller studies allowed us to estimate the extent of the association with more certainty than was possible within individual studies. We also saw factors that were examined repeatedly in multiple studies that were found not to significantly influence treatment outcomes even when combined.

We found that certain characteristics of migraine headaches, migraine history, and the presence of comorbid conditions such as depression or obesity can help predict the response to treatment with CGRP(-R) mAbs. Specific tests, such as quantitative sensory testing, transcranial doppler, or measurement of baseline salivary CGRP levels could also be useful in estimating the likelihood of benefiting from CGRP(-R) mAbs treatment, though this association needs to be validated by larger studies. Data on biomarkers or genetic markers and response rates to CGRP(-R) mAbs are lacking, which highlights a need for more studies that explore these factors.

Unilateral pain, accompanying photo-/phonophobia, and nausea with vomiting, which are listed as diagnostic criteria for migraine in the international classification of headache disorders (ICHD-3 [87]), were associated with a better response to CGRP(-R) mAbs. One hypothesis to explain this observation could be that CGRP plays a central role in mediating symptoms that are more typical of migraine. For example, CGRP plays a key role in modulating gastrointestinal tract motility, intestinal blood flow, and inflammation, and was proposed to account for the symptoms accompanying migraine, such as nausea, reflux, vomiting, and diarrhea [88]. Infusion of CGRP led to migraine-like photophobia in mice [89]. Headaches that have more of migraine-typical symptoms would then be better prevented by CGRP-targeted treatment than those that do not present with migraine-typical features.

The presence of central sensitization and its associated symptoms were particularly predictive of treatment failure. Peripheral and central sensitization leads to allodynia, first during migraine attacks, then, as the chronification of migraine progresses, even during interictal periods, through increased excitability of second and third-order neurons [90]. Ashina et al., hypothesize that in patients with interictal allodynia, central trigeminovascular neurons are sensitized and hyper-responsive, able to be activated independent of the activity of peripheral neurons, whereas in those without interictal allodynia, such a central sensitization has not yet taken place [43]. Interictal cephalic and extra-cephalic allodynia is likely to cause pain even in periods between attacks, and if sensitization continues, could lead to the occurrence of daily headaches [91]. Both interictal allodynia and daily headaches in chronic migraine were associated with failure of CGRP-targeted treatments. The presence of cutaneous allodynia is also linked with poor response to triptans, with a decreased likelihood of being pain-free 2 h after triptan treatment (15% vs. 93% in non-allodynic attacks) [92]. Nonresponse to triptans may partially be attributable to central sensitization, and this may explain why in our study, a good response to triptans was robustly associated with responding well to CGRP-targeted treatment.

Risk factors for migraine chronification such as obesity [93], depression [94], and medication overuse [95], were also linked with non-response to CGRP(-R) mAbs in our study, though to a smaller extent than interictal allodynia. Barbanti et al. [51] hypothesized that in obesity, which was shown to increase CGRP activity [96,97], the CGRP(-R) mAbs alone are insufficient to disrupt CGRP signaling. In a post hoc analysis of a randomized clinical trial, Martin et al. [98] found that subgroups of patients with obesity (obesity class I: BMI 30–35, and class II: BMI > 35) receiving a dose of eptinezumab of 100 mg did not have a ≥50% response rate superior to placebo (50%, 95% CI: 41–59% vs. 45%, 95% CI: 35.6–54.3% for class I obesity, and 37.1%, 95% CI: 25.1–49.1% vs. 34.5%, 95% CI: 24.5–44.5% for class II obesity). This was remedied by giving a higher dose (300 mg) of eptinezumab, which led to a treatment effect of more than 10% separation from placebo for patients with both class I and II obesity. In phase II and III randomized clinical trials of fremanezumab and galcanezumab, however, baseline weight or obesity, even with BMIs higher than 30 kg/m^2^, did not influence treatment outcomes [99,100], so it is uncertain whether weight or obesity are reliable predictive factors of response. Psychiatric comorbidities in real-world settings, especially comorbid depression, were associated with treatment nonresponse. One possible explanation is that increasing severity of migraine conveys a higher risk of developing comorbid depression and anxiety disorder [45], and the associations that were observed are attributable to higher migraine disease burden at baseline. Interestingly, migraine patients with depressive symptoms had higher salivary CGRP levels independent of MHD [46], indicating that for comorbid depression, there seems to be a separate and added pathology beyond migraine severity that may contribute to treatment failure.

Patients that had more failures with preventive medications in the past were less likely to be good responders. Medication overuse, which presupposes at least ten headache days per month severe enough to require medication, was also associated with a decreased likelihood of responding to CGRP(-R) mAbs. Overall, the further migraine patients advanced along the process of migraine chronification, the higher the chance seems to be of not profiting from CGRP-targeted treatment.

Several limitations exist that need to be considered when interpreting the results of this study. A significant portion of included studies were retrospective and are susceptible to information bias. Including only patients with analyzable data may lead to selection bias. Studies were often not explicitly designed to determine the predictive value of a specific factor; rather, an exploratory analysis was conducted, comparing all available variables and their association with response rate. Often, nonsignificant associations were not reported, possibly leading to publication or positive results bias. In our study, effect sizes from univariable and multivariable analyses were both included in the quantitative syntheses, potentially leading to pooled effect sizes inflated through confounding. However, as there are no hitherto established risk factors for non-response, there are no clear principles to guide which variables to include in a multivariable model. Often studies simply included all variables that had significant associations with the response rate in one multiple logistic regression analysis. Finally, causality cannot be established by this analysis.

There was high variability in the methods used among the included studies. Follow-up periods ranged from 3 months to a year. Different thresholds and either headache or migraine days were used to determine a response to treatment. Though most studies used the 50% reduction in MHD as the definition for a response, a few based the response on the change in MMD, set alternate thresholds, or did not dichotomize the cohort (Appendix A).

Despite these potential limitations and biases, this study offers a comprehensive synthesis of published studies on the predictors of response to CGRP(-R) mAbs, revealing certain trends that were consistent across the included studies.

Using results from our study, future research could focus on the predictive factors we identified to cement and quantify the relationship between the factor and treatment success, which then can be used to provide patients with individualized assessments of the likelihood of response. If indeed further progress along the continuum of migraine chronification makes a response to CGRP(-R) mAbs more unlikely, clinicians could influence the disease outcome by recommending CGRP(-R) mAbs earlier in the course of the disease before chronification and central sensitization occur. Better characterizing the subgroup of migraine patients that do not benefit from CGRP(-R) mAbs could help determine the mechanism behind treatment failure. Clarity about the mechanism of treatment failure will help in designing future therapies that could reverse central sensitization and migraine chronification, ultimately helping migraine patients who suffer the most.

## 4. Materials and Methods

We decided to perform a systematic search to identify all relevant studies regardless of predictive factor examined. As there are no previously published reviews on this topic, we performed a scoping review [101] with the goal of including as many relevant studies as possible to determine the nature and extent of published evidence and identify potential predictive factors. As such, quality assessments and evaluation of the risk of bias were not performed. Where appropriate, reported quantitative measures were synthesized to provide a more definite summary of available data and a clearer overview.

### 4.1. Literature Search

We searched Embase and MEDLINE databases on 16.04.2023 via Ovid and Pubmed using a combination of the search terms: CGRP(-R) mAbs, erenumab, fremanezumab, galcanezumab, or eptinezumab, migraine, predict, response, and responder (see also Appendix A). Published articles of real-world studies were included if: (a) they included patients with either EM or CM or both, with or without aura who received treatment with CGRP(-R) mAbs for at least three months, (b) the outcome of interest was response to CGRP(-R) mAbs defined as reduction in either MHD or MMD, either in absolute number or proportion reaching a threshold, and (c) if they provided a measure of association between possible predictive factors and response rate. If articles were based on overlapping cohorts, the one reporting more comprehensive data was included.

### 4.2. Data Extraction and Synthesis

From each included study, information on the cohort size, location, study type, CGRP(-R) mAbs given, proportion of EM and CM patients, inclusion/exclusion criteria, duration of follow-up, definition of endpoints, and predictive factors examined were extracted. Where available, percentage of patients with medication overuse, average number of previous preventive medication failures, 50% responder rate, and mean reduction in headache or migraine days were also extracted.

For quantitative synthesis, ORs from multivariable or univariable analysis and their 95% CI, preferably for ≥50% response in either MHD or MMD, were extracted from each study. If ORs were not reported and the factor of interest was a binary variable, we calculated univariable ORs from crude frequencies. If the same factor was examined for more than two extracted effect sizes, a quantitative synthesis was performed using the inverse variance method and random-effects model. Heterogeneity was assessed using the *I^2^* statistic [102]. R version 4.2.3 (“Shortstop Beagle”) [103], RStudio [104], and the meta and dmetar packages were used to calculate pooled effect sizes and generate forest plots [105,106].

Reported associations that were expressed in other ways than ORs, such as mean differences, medians with interquartile ranges, and proportions were summarized narratively.

### 4.3. Categories of Predictive Factors

Individual factors were grouped into categories and the results summarized for each category. Four overarching categories could be defined based on published associations thus far: (a) demographic characteristics, (b) migraine attack features, (c) migraine history, and (d) comorbid conditions. Factors categorized as demographic characteristics included age, sex, weight, employment status, level of education, and family history of migraine. Migraine attack features comprised laterality of pain (unilateral vs. bilateral) accompanying symptoms that included allodynia, and as a separate feature, interictal allodynia. Baseline migraine burden, number of previous preventive medications, acute medication use including response to triptans, and medication overuse were categorized under migraine history. Comorbid conditions were divided into psychiatric and non-psychiatric comorbid conditions.

## 5. Conclusions

Based on the currently available literature of mainly real-world studies with CGRP(-R) mAbs, response to triptans, unilateral pain with or without unilateral autonomic symptoms, and accompanying migrainous symptoms such as nausea, vomiting, photophobia, and phonophobia, were predictors of a good response (≥50% reduction in MMD or MHD) to CGRP(-R) mAbs. Obesity, interictal allodynia, the presence of daily headaches, a higher number of previous prophylactic medications, and psychiatric comorbidities including comorbid depression were predictive of a poor response (<50% reduction in MMD or MHD) to CGRP(-R) mAbs. Symptoms and signs of central sensitization and advanced migraine chronification seem to be associated with a poor response to CGRP(-R) mAbs. As more and more migraine patients are being treated with CGRP(-R) mAbs, further real-world studies should investigate whether these associations can be replicated or identify additional significant predictors of response.

## Figures and Tables

**Figure 1 pharmaceuticals-16-00934-f001:**
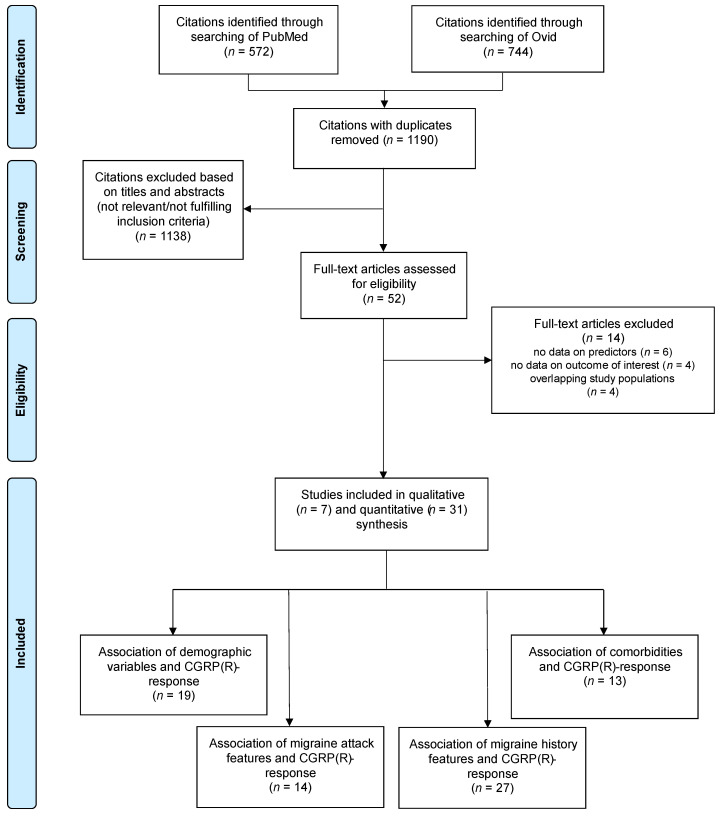
PRISMA flow diagram of included studies. Some studies reported associations on multiple categories of predictors.

**Figure 2 pharmaceuticals-16-00934-f002:**
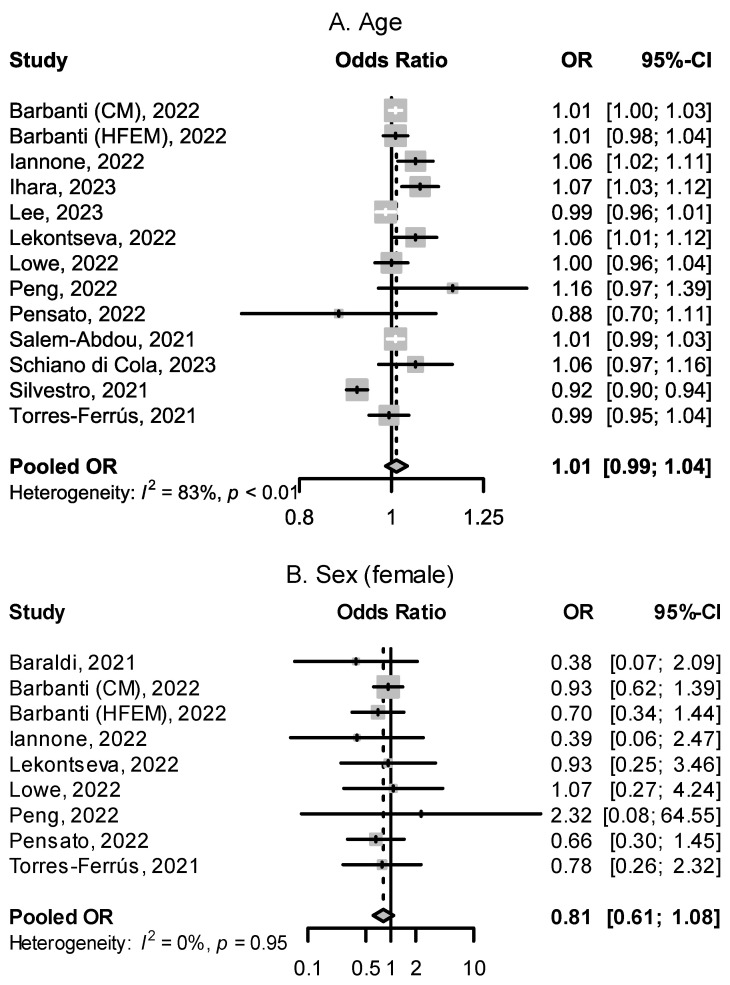
Effect of age and sex on treatment response. (**A**): ORs for age in years for the outcome of treatment success. (**B**): ORs for female vs. male sex for the outcome of treatment success. HFEM: high-frequency episodic migraine, and CM: chronic migraine [44,50,51,56,58,59,60,61,62,65,66,67,68].

**Figure 3 pharmaceuticals-16-00934-f003:**
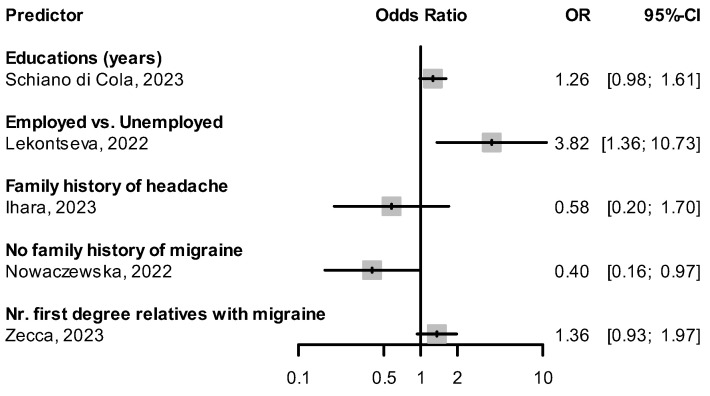
Effect of additional demographic baseline variables on treatment response [42,56,57,59,62].

**Figure 4 pharmaceuticals-16-00934-f004:**
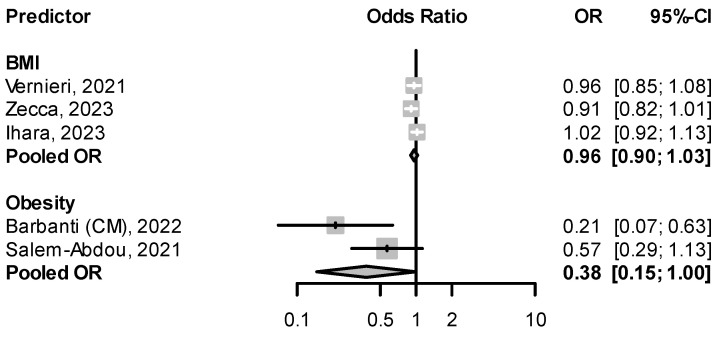
Effect of weight on treatment response BMI: body mass index in kg/m^2^ [48,51,56,57,66].

**Figure 5 pharmaceuticals-16-00934-f005:**
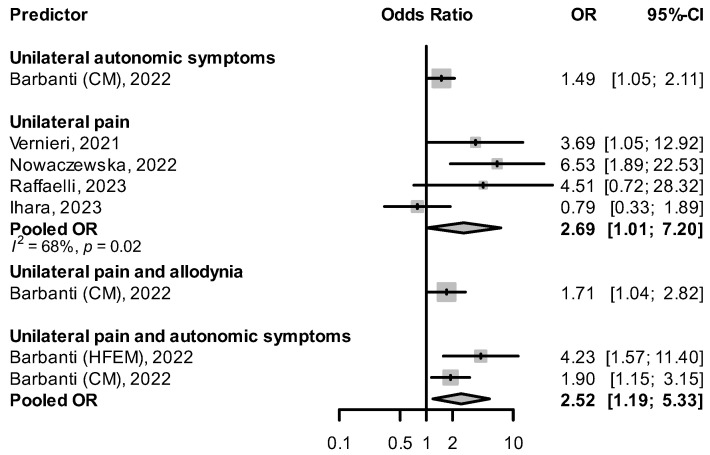
Effect of the presence of unilateral symptoms on treatment response HFEM: high-frequency episodic migraine, and CM: chronic migraine [42,48,51,55,56].

**Figure 6 pharmaceuticals-16-00934-f006:**
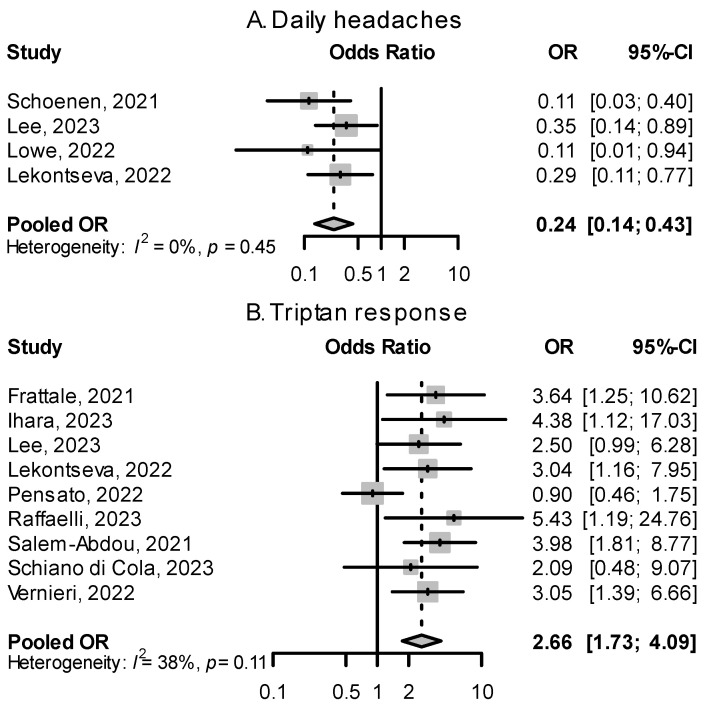
Effects of having daily headaches and being triptan responders on treatment response. (**A**): ORs for those with daily headaches (continuous chronic migraine) vs. noncontinuous headaches for the outcome of treatment success. (**B**): ORs for triptan responders vs. triptan non-responders for the outcome of treatment success [52,53,55,56,59,60,61,62,64,66,68].

**Figure 7 pharmaceuticals-16-00934-f007:**
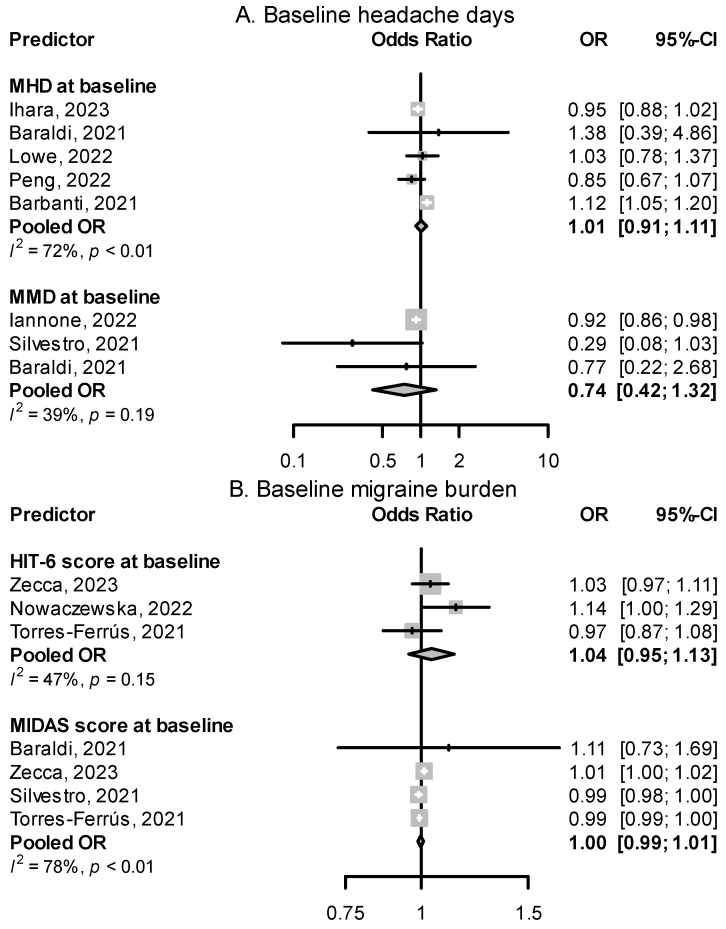
Effect of baseline headache days (**A**) and measures of headache disability (**B**) on treatment response. MHD: monthly headache days, MMD: monthly migraine days, MIDAS: migraine disability assessment, and HIT-6: headache impact test-6 [42,44,50,56,57,58,61,65,67,74].

**Figure 8 pharmaceuticals-16-00934-f008:**
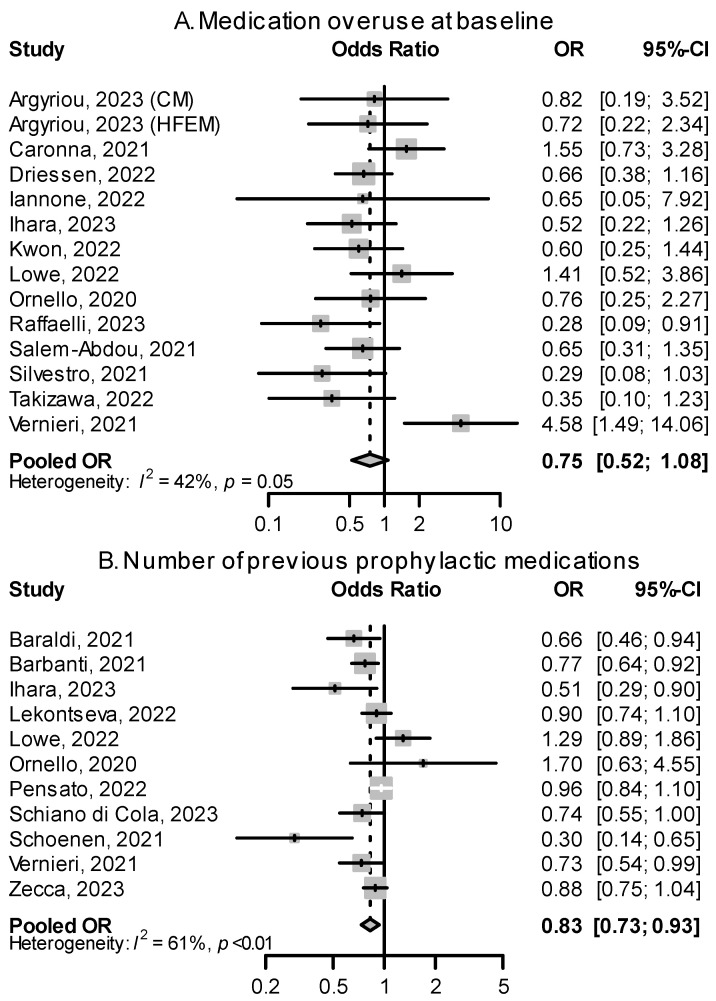
Effects of medication overuse status and the number of previous prophylactic medications on treatment response (**A**): ORs for patients with medication overuse vs. no medication overuse at baseline for the outcome of treatment success. (**B**): ORs showing the effect of each added previous prophylactic failure on the outcome of treatment success. HFEM: high-frequency episodic migraine. CM: chronic migraine [47,48,50,53,54,55,56,57,58,59,61,62,63,65,66,68,69,74,75,78].

**Figure 9 pharmaceuticals-16-00934-f009:**
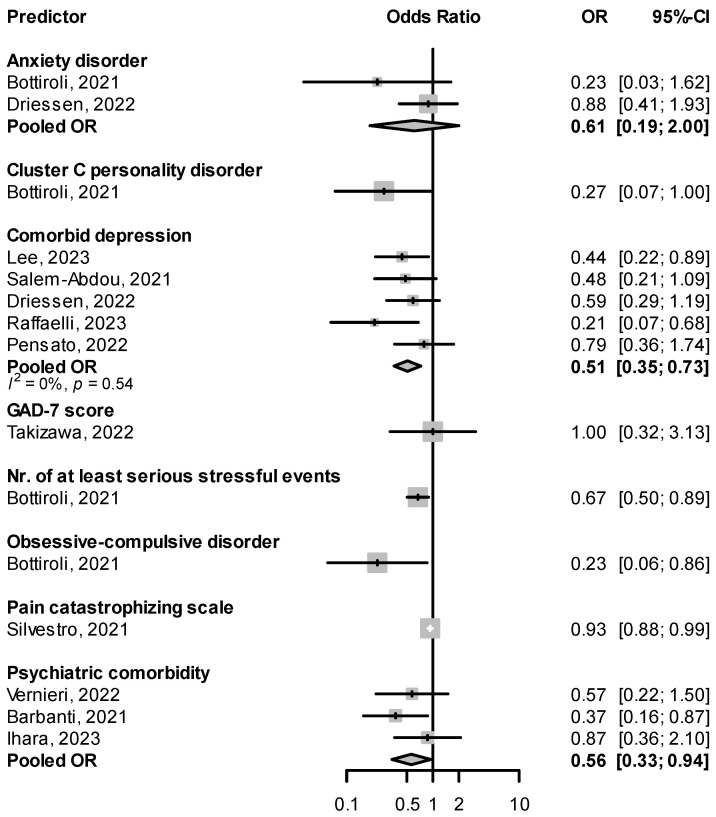
Effect of comorbid psychiatric disorders or measures of psychological vulnerability at baseline on treatment response. GAD-7: generalized anxiety disorder-7 [45,47,55,56,58,60,64,66,68,74,78].

## Data Availability

The datasets used and/or analyzed during the current study are available from the corresponding author upon reasonable request.

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
