# Peer review of "A Scoping Review and Meta-Analysis of Anti-CGRP Monoclonal Antibodies: Predicting Response"

_pharmaceuticals, 2023, doi:10.3390/ph16070934_

Round 1

Reviewer 1 Report

The paper by Hong et al. summarizes and provides a meta-analysis of all predictors potentially involved in achieving a response status (usually a 50% reduction in MHD/MMD) for anti-CGRP/R mAbs in patients with migraine. The data are derived from numerous real-world studies on anti-CGRP/R mAbs published so far.

The paper is well-structured and easy to read, making it potentially very useful for researchers and clinicians. Moreover, it is timely, considering the growing amount of research on the topic and the need to provide an overview of what has been done on predictors to date. Indeed, several and very different predictors have been reported with elevated inconsistency among studies, and no strong predictor has emerged so far.

I have just one major comment and some minor comments and suggestions.

Major comment:

Several studies have been published by the same large group of Researchers who conducted different multicenter studies in Italy (Barbanti et al., Vernieri et al.), mainly on erenumab (EARLY) and galcanezumab (GARLIT). Although it is clear that EARLY and GARLIT reported different cohorts of patients, the paper by Barbanti et al. (doi:10.1186/s10194-022-01498-6) on predictors, which includes more than 800 patients during the same recruitment period as EARLY and GARLIT, may have a considerable overlap of patients from these studies. This could introduce selection bias considering the non-independence of the cohorts included in the meta-analysis.

Minor comments:

In the abstract, I suggest specifying that the response to CGRP is based on a 30%/50% reduction in MHD/MMDs at different follow-up intervals. Furthermore, it should be explained that data on predictors through studies are inconsistent and sparse so far.

Page 6 "Weight": It could be useful to specify the "cut-off" used to define obesity in these studies. Additionally, was BMI stratified as a categorical variable?

Page 11 "Psychiatric comorbidities": Please briefly specify how psychiatric comorbidities were assessed in these studies, if at all. Usually, no diagnosis of major depression or anxiety disorders was reported according to DSM-5, but unspecific, non-specialist diagnoses are included in these real-world studies. Only a few studies (especially Bottiroli et al., 2021) carefully reported psychological traits and psychiatric comorbidities assessed by a psychologist/psychiatrist.

A Bayesian reporting of research terms (Search strings) could be useful for reproducibility. It could be added as a supplementary material.

Several references/figure indexings are not available in the text due to some errors ("Error! Reference source not found"). Please check them carefully. Furthermore, some studies reported in the meta-analysis are not listed in the main reference list (although they are reported in the supplementary material). Please provide the full list of references in the main text to better identify the studies reported in the meta-analysis.

Figure 1: The four PRIMA categories on the left (identification, screening, etc.) are not aligned with the flowchart. Please modify accordingly.

According to ILAE, the term "Antiepileptics" should be modified to "anticonvulsants/antiseizure medications."

Page 8: The legend for Figure 7 is erroneously reported in the main text.

Reviewer 2 Report

This is a valuable and thoroughly researched review. It addresses a major practical question and provides some initial answers which will, no doubt, be confirmed or modified by further studies. The forest plot figures displaying odds ratios are clear and helpful.

The discussion is well balanced, with some welcome speculation as to the mechanisms by which various factors may be contributing to CGRP Ab responsiveness. The limitations of the study are discussed appropriately.

Reviewer 3 Report

The review report on the manuscript, titled Predictors of response to anti-CGRP monoclonal antibodies: a scoping review and meta-analysis of real-world experience” by Hong JB et al., submitted to Pharmaceuticals

 Understanding the underlying mechanisms and predictors of treatment response is necessary to the development of more efficient and individualized treatments for migraines. In this manuscript, entitled ‘Predictors of response to anti-CGRP monoclonal antibodies: a scoping review and meta-analysis of real-world experience’, Hong and colleagues explore real-world experiences to identify factors that can help predict the effectiveness of anti-CGRP monoclonal antibodies.

This manuscript's main strength is that it addresses a timely and fascinating topic, presents a thorough scoping review and meta-analysis of real-world experiences, and offers insightful information about the variables that can help predict the efficacy of these antibodies for the treatment of migraines.

In general, I think the idea of this review article is interesting, and the author’s fascinating observations on this timely topic may be of interest to the readership of Pharmaceuticals. However, some comments as well as some crucial evidence should be included to support the author’s argumentation and improve its adequacy, readability, and thus the quality of the manuscript prior to publication. My overall opinion is to publish this research article after the authors have carefully considered my comments below for the betterment of the manuscript.

Please consider the following comments:

1.   First, I would like the authors to make sure the presence of all elements for a scoping review and a meta-analysis by using the checklist [1-3] and to follow the journal’s guidelines focusing on the structures of articles (Introduction, Results, Discussion, Methods, and Conclusion) [4].

2.   Title: Please present a concise and self-explanatory title stating the most important findings of this review. Suggestions: "Real-World Experience: Predicting Response to Anti-CGRP Monoclonal Antibodies"; "A Scoping Review and Meta-Analysis of Anti-CGRP Monoclonal Antibodies: Predicting Response"; "The Key to Success: Identifying Predictors of Response to Anti-CGRP Monoclonal Antibodies" [5-7].

3.   Abstract: Please expand the abstract to 200 words according to the guidelines of the journal [4], proportionally presenting the background, the methods, the results, and the conclusion. The background should include the general background (one to two sentences), the specific background (two to three sentences), and current issue addressed to this study (one sentence), leading to the objectives. In this subsection I would like the authors to lay out basic information, problem statement, and the authors’ motivation to break off.

The methods should clarify the authors’ approach such as study design and variables to solve the problem and/or to make progress on the problem. The results section must state the results in numbers and clarify their statistical significance. This subsection should close with one to two sentences which put the results into a more general context. The conclusion should open with one sentence describing the main result using such words like “Here we show”, which should be followed by statements such as the potential and the advance this study has provided in the field and finally a broader perspective (two to three sentences) readily comprehensible to a scientist in any discipline [8-11].

4.   Keywords: Please list ten keywords chosen from Medical Subject Headings (MeSH) [12] and use as many as possible in the title and in the first two sentences of the abstract.

5.   A graphical abstract is highly recommended.

6.   Introduction: The authors need to fully expand this section with about 1000 words and several paragraphs, introducing information on the main constructs of this study, which should be understood to a reader in any discipline and make persuasive enough to put forward the main purpose of current research the author has conducted and the specific purpose the author has intended by this review. I would like to encourage the authors to present the introduction starting with the general background, proceeding to the specific background, and finally the current issue addressed to this study, leading to the objectives. Those main structures should be organized in a logical and cohesive manner [13,14]

7.   In this regard, the following works may enhance the value of this manuscript, including but not limited to: https://doi.org/10.3390/cells11213498; https://doi.org/10.3390/cells11193092; https://doi.org/10.3390/cells11142258;  https://doi.org/10.3390/cells11111768; https://doi.org/10.3390/cells12010; https://doi.org/10.3390/cells11111768143; https://doi.org/10.3390/cells11172767; https://doi.org/10.3390/cells11152444; https://doi.org/10.3390/cells11152406.

8.   Methods: I recommend opening this section with a short introductory paragraph and cite more references to ensure the reliability and the integrity of evidence in the study design the authors build and the methodology they have decided to apply.

9.   Results: I would like the authors to close this section with a paragraph which puts the results into a more general context.  

10.            Discussion: I would like the authors to reorganize this section by opening with an introductory paragraph and followed by the summary of the previous section (Results). Then, I expect the authors to develop arguments clarifying the potential of this study complementing as the extension of the previous work, the implication of the findings of this study, how this study could facilitate future research, the ultimate goal, the challenge, the knowledge and the technology necessary to achieve this goal, the statement about this field in general, and finally the importance of this line of research. It is particularly important to present its merit, and its potential translation of this study to clinical practice [15].

11.     Conclusion: I think that presenting the conclusion would benefit from a single paragraph presenting some thoughtful as well as in-depth considerations by the authors as experts to convey the take-home message. The authors should make an effort to explain the theoretical implications as well as the translational application of their research. I believe that it would be necessary to discuss theoretical and methodological avenues in need of refinement as well as suggestions for a path forward in understanding the importance of this study.

References: Please follow the guidelines of the journal (https://www.mdpi.com/journal/biomedicines/instructions). Journal abbreviation and page number should be ended with period. I would like the authors to cite more references. Typically, a review article like this includes over 150 references.

Overall, the manuscript contains nine figures, supplementary tables, and 58 references. I believe that the manuscript may have important value in providing evidence-based insights into the effectiveness of these antibodies for migraine treatment, which can inform clinical decision-making and improve patient outcomes. I hope that, after careful revisions, the manuscript can meet the journal’s high standards for publication.

References:

  1. https://onlinelibrary.wiley.com/doi/full/10.1111/j.1471-1842.2009.00848.x
  2. https://www.edanz.com/excited-science/systematic-scoping-narrative
  3. https://link.springer.com/chapter/10.1007/978-981-16-5248-6_29

4.     https://www.mdpi.com/journal/pharmaceuticals/instructions

5.     https://plos.org/resource/how-to-write-a-great-title/

6.     https://www.nature.com/nature-index/news-blog/how-to-write-a-good-research-science-academic-paper-title

7.     https://www.indeed.com/career-advice/career-development/catchy-title

8.     https://www.scribbr.com/dissertation/abstract/

9.     https://writing.wisc.edu/handbook/assignments/writing-an-abstract-for-your-research-paper/

10.  https://doi.org/10.5812/ijem.100159

11.  https://doi.org/10.4103/sja.SJA_685_18

12.  https://meshb.nlm.nih.gov/

13.  https://dept.writing.wisc.edu/wac/writing-an-introduction-for-a-scientific-paper/

14.  https://doi.org/10.3163/1536-5050.103.2.001

15.   https://www.scribbr.com/dissertation/discussion/    

Minor editing

Reviewer 4 Report

aCGRP treatment for migraine prevention seems highly effective and offers a new pharmaceutical target treating the disorder.

This therapy is more expensive, thus it might be restricted to a limited population of migraineurs, therefore it is important to know the predictors of success/failure which are summarized in the manuscript.

The paper is straightforward and important from the point of view of the clinicians.

Minor remarks:

The authors should add data about the effectivity in menstruation related migraine as well.

I don’t see anyone for affiliation 2 from the authors list.

Round 2

Reviewer 3 Report

No